# Doping of Transparent Electrode Based on Oriented Networks of Nickel in Poly(3,4-Ethylenedioxythiophene) Polystyrene Sulfonate Matrix with P-Toluenesulfonic Acid

**DOI:** 10.3390/nano13050831

**Published:** 2023-02-23

**Authors:** Irek R. Nizameev, Guliya R. Nizameeva, Marsil K. Kadirov

**Affiliations:** 1Arbuzov Institute of Organic and Physical Chemistry, FRC Kazan Scientific Center, Russian Academy of Sciences, Arbuzov Str. 8, Kazan 420088, Russia; 2Department of Nanotechnology in Electronics, Kazan National Research Technical University named after A.N. Tupolev—KAI, 10, K. Marx Str., Kazan 420111, Russia; 3Department of Physics, Kazan National Research Technological University, 68, K. Marx Str., Kazan 420015, Russia

**Keywords:** nanowires, nanonetworks, conductive coating, nickel, transparency, composite material, doping, p-toluenesulfonic acid, TCE

## Abstract

This work aimed to obtain an optically transparent electrode based on the oriented nanonetworks of nickel in poly(3,4-ethylenedioxythiophene) polystyrene sulfonate matrix. Optically transparent electrodes are used in many modern devices. Therefore, the search for new inexpensive and environmentally friendly materials for them remains an urgent task. We have previously developed a material for optically transparent electrodes based on oriented platinum nanonetworks. This technique was upgraded to obtain a cheaper option from oriented nickel networks. The study was carried out to find the optimal electrical conductivity and optical transparency values of the developed coating, and the dependence of these values on the amount of nickel used was investigated. The figure of merit (FoM) was used as a criterion for the quality of the material in terms of finding the optimal characteristics. It was shown that doping PEDOT: PSS with p-toluenesulfonic acid in the design of an optically transparent electroconductive composite coating based on oriented nickel networks in a polymer matrix is expedient. It was found that the addition of p-toluenesulfonic acid to an aqueous dispersion of PEDOT: PSS with a concentration of 0.5% led to an eight-fold decrease in the surface resistance of the resulting coating.

## 1. Introduction

Transparent electrodes (TCEs) are a special class of electrodes due to their optical density and selectivity. They are the main components of many optoelectronic devices such as solar panels, organic light-emitting diodes (OLEDs), liquid crystal displays (LCDs), transparent heaters, and smart windows. Indium tin oxide (ITO) has excellent optoelectronic properties and environmental stability [1]. Commercial ITO thin films typically have a sheet resistivity of 20 ohm/sq for films 100–300 nm thick. Their transparency coefficient is more than 80% in the visible range of the spectrum. ITO is the dominant material in optoelectronics, but this oxide has several significant drawbacks [2,3,4]. First, the rising cost of indium is associated with the lack of world resources and its large consumption [5,6,7]. The cost also depends on the deposition methods and technological progress [8]. Typically, most commercial ITO thin films are deposited using magnetron sputtering, molecular beam epitaxy, thermal evaporation, and pulsed laser deposition. All of these methods require deposition processes at temperatures of 400 to 500°C or higher and sophisticated tools for high-vacuum deposition [9,10]. This makes ITO unsuitable for use in flexible electronics. Many flexible substrates break down at these high temperatures. In addition, the inefficient use of the material in the deposition process is another factor that increases the cost of ITO. Therefore, there is a huge demand for the development of new electrode materials with lower costs and comparable characteristics to ITO.

Over the past few years, various alternative transparent conductive materials have been developed to address the above ITO problems. These are doped metal oxides [11], transparent conductive polymers, carbon nanomaterials, and metal nanofilms and nanowires. Some of the known binary oxides are based on cadmium oxide, for example, cadmium oxide doped with indium (In) [12], tin (Sn) [13], fluorine (F) [14], or yttrium (Y) [15]. Ternary compounds are also known, such as CdSnO_3_, Cd_2_SnO_4_, and CdIn_2_O_4_ [12,16].

There are also works describing binary oxides such as In_2_O_3_:Ti [17], In_2_O_3_:Zr [18], In_2_O_3_:Nb [19], and In_2_O_3_:W [20].

A promising alternative to ITO is zinc oxide (ZnO). This oxide is an n-type semiconductor with a band gap of 3.28 eV. The concentration of charge carriers is approximately 10^17^ cm^−3^ [21]. However, ZnO, despite its good optical transparency, has low electrical conductivity. To increase the conductivity, this oxide is doped with various elements. For example, the production of transparent conductive thin films of multicomponent oxide ZnO-In_2_O_3_ using various deposition methods is reported [22,23]. However, it should be noted that the conductivity of the resulting coating strongly depends on the amount of indium, the deposition method, and the temperature. Authors have obtained films with various amounts of indium.

Another interesting direction in the creation of transparent electrodes is the development of zinc oxide nanostructures. Zinc oxide nanostructures exist in various morphologies, such as nanowires, nanorods [24,25], nanotubes, and nanoflowers [26]. Various physical and chemical processes are used to obtain them. ZnO nanoflowers are a unique three-dimensional structure, which is also promising material as photoanodes in solar cells. Jiang et al. made a dye-sensitized solar cell using nanoflowers [26]. According to the authors, the photoanode in the form of a nanoflower contributes to the efficient loading of the dye and the collection of light, and it has high electronic conductivity. The nanoflower cell demonstrated a photoconversion efficiency of 1.9%, while the solar cell efficiency using ZnO nanorods was 1.0%.

Carbon nanomaterials can also be used in transparent electrodes [27]. Many studies report on graphene-based hybrid materials [28,29,30,31]. Despite numerous studies of graphene as a transparent electrode, it is worth noting that the production of high-quality graphene is only possible via mechanical splitting or CVD. However, to date, none of these methods has been considered suitable for creating low-cost photovoltaic cells [6,32]. Very often, graphene is used not as an independent electrode but as an alloying additive [33,34].

Due to their combination of transparency and conductivity, conductive polymers are considered a good alternative to ITO [35]. The most common and most studied conductive polymers to date are polyaniline (PANI), polypyrrole (PPY), and poly(3,4-ethylenedioxythiophene):polystyrene sulfonate (PEDOT:PSS). By themselves, these polymers have a high surface resistance and, accordingly, low conductivity. Therefore, to increase their conductivity, researchers suggest using various additives or creating composite materials with metal nanowires, carbon nanotubes, graphene, etc. [36,37,38,39,40,41,42,43]. An electrode based on Ag-PEDOT:PSS (polymer + silver) was developed and used in organic light-emitting diodes as an anode [44]. Transparent conductive polymers may be better alternatives to ITO because they are readily available and allow flexible films to be made. However, these polymers, when used alone, especially at high temperatures, remain unstable in an air environment. The surface resistance of polymer electrodes in an air environment increases with time [45,46]. However, when used with other materials, conductive polymers show good results, so they require further research.

Among all alternative ITO materials, transparent metal-based electrodes are considered to be the most promising. They are the best candidates due to their high electrical conductivity, the ability to achieve high optical transparency through the use of nanostructures, and the ability to control properties such as flexibility and extensibility through doping. The best electrically conductive material at room temperature is silver (Ag), with an electrical conductivity of 6.3 × 10^7^ S/m. Accordingly, Ag-based nanomaterials have good potential as efficient transparent electrodes. Along with silver, which is the subject of a large number of studies [47,48,49,50,51,52,53], other metals such as Cu [54,55,56], Ni–Cu [57,58], and Au alloys have recently been studied. Madaria et al. have developed transparent electrodes based on silver nanonetworks with optical transparency of 85% and surface resistance of 33 ohm/sq. Silver nanonetworks (AgNWs) were synthesized in solutions and transferred onto substrates [59]. However, the process of transferring metal nanonetworks from solutions to appropriate substrates is a significant problem in the manufacture of transparent electrodes. During the transfer process, most of the nanonetworks are lost, destroyed, and cracked [60,61]. In 2020, another group of researchers synthesized AgNWs with a length of 130 μm and an aspect number of 1000. The synthesized nanowires were deposited on a glass substrate through sputtering. The result was transparent electrodes with a transparency coefficient of 91% and a surface resistance of 4.6 Ω/sq [62]. To obtain electrodes based on AgNWs, the spin-coating method is very often used [63,64,65]. The spin-coating method makes it possible to obtain uniform films of nanowires on various surfaces.

Cheng et al. developed transparent electrodes via drip casting. To do this, they synthesized AgNWs with a large aspect number [66]. The finished AgNWs were cast onto patterned glass substrates. Thus, a homogeneous conducting network was formed on the glass substrate. In [67], the authors developed transparent electrodes through vacuum filtration. Compared with vacuum filtration, the deposition of metal nanowires using Mayer rods is considered to be simpler and more scalable [68]. For example, transparent electrodes based on AgNWs with a surface resistance of 21 Ω/sq and transparency of 95% on a surface of 20 × 20 cm^2^ were obtained using this method [69]. Yang et al. made AgNW-PEDOT:PSS transparent composite electrodes. The authors used a conducting polymer to reduce the contact resistance between the individual nanowires.

In the manufacture of transparent electrodes, in addition to silver nanowires, copper nanowires are also used [54,55,70,71,72,73]. However, the developed electrodes are not stable in the air. To prevent oxidation, a network of metallic copper nanowires is embedded on the surface of transparent fiberglass [72], or a CuNW-PEDOT:PSS metal–polymer composite electrode is created [74]. Considering the high cost of gold, AuNW gold nanowires at first glance seem less attractive than copper and silver nanowires. However, due to their unique mechanical and optical properties and high chemical stability, interest in gold nanowires is only growing, and, along with copper and silver nanowires, they are promising materials for creating transparent electrodes [75,76,77,78,79,80]. The self-assembly of AuNWs on substrates without the use of a template is reported. A honeycomb microporous pattern of gold nanowires was formed on the surface of the transparent substrate. The surface resistance of an electrode based on such a nanostructure was 125 Ω/sq. In some modern works, even silicon materials are used. The authors of [81] demonstrated the use of a combination of electroactive self-assembly and coordination chemistry for the synthesis of ordered and vertically oriented thin films of mesoporous silicon dioxide containing covalently bonded chromophores. The developed material has potential in optoelectronic applications.

In this research, we studied nickel networks with modified PEDOT:PSS as a transparent electrode. A thin coating was created on the surface of a glass substrate according to a technique developed earlier and described in [82]. In general, this work is a logical continuation of a previous investigation [82], with an improved method and detailed research on the influence of process parameters on the synthesized material.

## 2. Materials and Methods

### 2.1. Material Synthesis

Nickel nanowires on the glass surface were synthesized in two steps involving the preparation of surfactant self-assembly at the glass/surfactant solution interface and the subsequent reduction of metal precursors [82]. A drop of 1 mM cetyltrimethylammonium bromide (CTAB) and 0.1 mM nickel chloride (NiCl_2_·6H_2_O) water solution was applied on the surface of the glass and then kept for 30 min for the formation of surface micelles at the interface at the room temperature. The metal precursors were reduced with fifty-fold excess hydrazine hydrate for 30 min. The glass substrate with the applied drop was placed in a magnetic field (H = 400 mT) during the entire reduction process. The field was set in such a way that its magnetic field lines would be directed parallel to the plane of the glass substrate. After solvent evaporation, the surface of the substrate was cleaned with water and ethanol to remove the adsorbed surfactant micelles and dried at 80 °C. All reagents were purchased from Sigma-Aldrich (St. Louis, MO, USA).

The method for the synthesis of submicron nickel fibers: nickel chloride NiCl_2_∙6H_2_O (47.6 mg) was dissolved into 2 mL ethylene glycol (EG) under continuous stirring at room temperature for 30 min. Then, 8 mL EG solution of sodium hydroxide (m(NaOH) = 480 mg) and 6 mL EG solution of hydrazine hydrate (N_2_H_4_∙H_2_O) (4 mL, 85 wt.%) were added dropwise into the above solution containing Ni^2+^ ions. This reaction mixture was continuously stirred to obtain a homogeneous solution. The solution was then placed in a magnetic field and heated to 70 °C (the magnetic field strength was 400 mT). The reaction was completed after about five hours. Grey-black solid aggregates were adsorbed on the inner surface of the container in which the reaction took place. The aggregates were separated from the solution using a magnetic field and washed five times with distilled water and ethanol. Then, the products were dried at 70 °C for three hours.

### 2.2. Material Characterization

The morphology of the surfaces was investigated using an atomic force microscope (AFM) MultiMode V (RTESP Veeco (New York, NY, USA) cantilevers were used in the experiments). The AFM image resolution was 512 × 512 points, and the scanning rate was 1 Hz. The roughness (R_a_) of the samples was calculated as the arithmetic mean value of the profile deviations absolute values within the base length.

Transmission electron microscopy (TEM), selected-area electron diffraction (SAED), and TEM-EDX experiments were performed on a Hitachi (Tokyo, Japan) HT7700 (wolfram filament, accelerating voltage = 100 kV, Thermo Scientific (Waltham, MA, USA) energy-dispersive X-ray detector). The SEM studies were carried out using Tescan VEGA (Brno, Czech Republic) (accelerating voltage = 30 kV, time of collection = 300 s, work distance = 9 mm, and resolution = 1024 × 768). A Leica DCM 3D (Wetzlar, Germany) system was used to carry out optical microscopy imaging.

A current–voltage characteristic of the layer with Ni networks was measured using potentiostat–galvanostat P-20X (electrochemical instruments, Chernogolovka, Russia) in a four-electrode mode using four-probe gold electrodes. The sheet resistivity of the layer was determined using the standard van der Pauw method. The optical transparency of the coating was studied on a Specord 50 PLUS spectrophotometer (Analytik Jena AG, Jena, Germany).

The investigation of the temperature dependence of the developed coating surface resistance was carried out by placing the entire measuring part of the four-probe measuring system together with the sample in a thermal insulation chamber. Temperature regulation and stabilization in the chamber were carried out using a special system (Appendix A). This system was built based on a temperature stabilization unit B-VT-1000 from Bruker (Billerica, MA, USA), Germany. The system, in addition to the main one, included an additional digital temperature meter, namely the Digi-Sense (Orem, UT, USA) Thermocouple Thermometer. The temperature in the chamber was maintained with a stream of gaseous nitrogen coming through a vacuum tube from a Dewar vessel with liquid nitrogen. Inside the tube, there was an electric heater that regulated the temperature of the nitrogen gas. The B-VT-1000 control unit allows one to set the required temperature in the chamber by regulating the gas flow and voltage on the electric heater. The feedback procedure during temperature stabilization was carried out through a thermocouple connected to the control unit. To improve measurement accuracy, the temperature in the chamber was controlled at two points. The current–voltage characteristics of the test sample were recorded after the required temperature was set and stabilized.

## 3. Results and Discussion

In previous works, we have shown a method for creating a transparent electrically conductive coating based on the oriented networks of platinum and nickel [82,83,84]. This study demonstrates the optimization of this technique and its improvement in order to obtain the best combination of optical transparency and electrical conductivity of the layer. The numerical value (“quality index”) of the figure of merit (FoM) [85] is used in the literature to assess the quality of coatings (based on the indicators of transparency and surface resistance of the material):(1)$$\Phi_{TC}=\frac{{{(T}_{550nm})}^{10}}{R_{S}}$$

In addition to Expression (1), in the modern literature, there is another equation for determining the FoM of transparent electrodes, based on the relationship between $\sigma_{dc}$ and $\sigma_{opt}$, where $\sigma_{dc}$ is the conductivity under direct current, and $\sigma_{opt}$ is the optical conductivity at a wavelength of 550 nm [86]:(2)$$\frac{\sigma_{dc}}{\sigma_{opt}}=\frac{188.5}{R_{S}{{(T}_{550}}^{-0.5}-1)}$$

Both expressions were used in this work.

Our method is based on the chemical reduction of nickel salt. The shaping and orientation of the resulting system were carried out using a micellar template of a surfactant and a magnetic field. The use of a CTAB micellar template leads to the formation of metal nanonetworks with a domain structure of orientation, i.e., the orientation of the network changes from one domain to another [82]. Domain sizes ranged from one hundred microns to a millimeter in transverse dimension. The sizes of domains were presumably determined through the formation of a micellar template at the water–glass interface. Nickel is an inexpensive metal that is magnetized. This phenomenon can be used to control its shape and direction of growth. We applied a homogeneous magnetic field to the area of nanonetwork synthesis so that its field lines were parallel to the glass surface. The use of an external magnetic field made it possible to obtain a long-range order orientation and to single out the preferred direction of the domain orientation as a whole. Submicron nickel fibers were used to impart unity to the plurality of individual-oriented nickel nanonetworks. The result was a single conductive coating on the surface of the glass, which had transparency in the optical range.

The concentration of nickel used on the glass surface plays a key role. In all our works on the synthesis of metal nanonetworks (platinum, palladium, iron, cobalt, and nickel) [82,87,88,89,90,91] networks is possible only at a certain concentration of metal. This means that it is possible to change the values of the surface resistance and transparency coefficient for the developed coating only by changing the number of submicron nickel fibers. The amount of nickel in the nanonetwork remained fixed in all the experiments and corresponded to a concentration of 0.1 mM in the reaction mixture, which was deposited on a glass substrate. As a numerical parameter determining the amount of nickel deposited on the glass surface, we used the value of pure metal’s mass per surface area of the glass substrate. Thus, samples of the developed coating on glass with different amounts of metallic nickel were obtained: 0.6∙10^−6^–311∙10^−6^ g/cm^2^. The smallest value in this range corresponded to a coating consisting only of nickel nanonetworks. The most characteristic form of the obtained oriented networks for some values of the nickel deposition density is shown in Figure 1.

In the chemical reaction of obtaining nanonetworks of nickel and its submicron fibers, nickel salt was assumed to be fully reduced. This means that the resulting value of the deposited mass of metal could be determined from the salt mass in the initial solution. We studied the different stages of the reaction to determine the time during which all the nickel in the solution was reduced (a detailed description of the technique is given in the Appendix A).

It can be seen from the microscopic images that the declared nickel density range was extensive and covered the area from a very light coating to a very dense one. All the obtained samples were examined on a spectrophotometer in the 290–1100 nm wavelength range. The averaged transmission spectra for some nickel network deposition densities are shown in Appendix A. As expected, with an increase in the density of nickel deposition on the glass surface, a monotonous decrease in the transparency coefficient was observed over the entire optical range. With an increase in the surface density of nickel on glass, a decrease in the total area of uncoated (transparent) areas was observed (Figure 1). Therefore, one should expect a decrease in the transparency coefficient in the entire optical range for denser coatings compared with less dense ones. The transmission spectra of oriented nickel networks on a glass substrate in the UV, visible, and near-IR regions for different amounts of the deposited metal are shown in Appendix A. The dependence of the transparency coefficient at a wavelength of 550 nm on the surface density of the nickel network is shown in Figure 2A. The monotonic decrease in T550 (m/S) was well approximated using an exponential function (black line in Figure 2A).

In the course of the research, an interesting feature of the coating was discovered: In the range of 950 nm and above, a downward “bend” of the transmission spectrum was observed. This bend (increased NIR absorption) was observed only in the presence of nickel nanonetworks and was not characteristic of a pure submicron network. This phenomenon was discussed in [92]. The phenomenon can be related to the peculiarities of radiation transmission through subwavelength apertures. Bethe’s criterion shows that when the aperture size is less than half the wavelength of the radiation passing through it, its strong attenuation is already observed in the near field. In our case, this means that the material’s absorption coefficient increased when the critical value of the wavelength was reached. It should be noted that such an effect is possible only for regular structures, in which the spread of transverse dimensions is small. We believe that the discovered effect may be of interest for applications in the development of nanosensors and near-infrared sensors.

The next stage in the study of the developed coating was an investigation of the electrical conductivity dependence at various densities of nickel network deposition. However, before that, the developed technique required the deposition of a polymer layer on the surface of a glass substrate coated with an oriented nanonetwork and submicron nickel fibers. The application of a polymer layer provided uniformity and increased the adhesion of the layer to the substrate and the stability of properties. We chose poly(3,4-ethylenedioxythiophene) polystyrene sulfonate (PEDOT: PSS) as a polymer, which is easily applied from an aqueous dispersion. To create a thin uniform layer, we used the centrifugation method. Most devices for applying thin films to solid surfaces via centrifugation (spin coaters) use vacuum fixation of the sample. This method was not acceptable for our tasks because there was a high probability of contamination of the back side of the substrate with the particles of vacuum oil. Thus, we used a system with a removable cartridge holder (Ossila company, Appendix A).

The results of surface resistance measurements for coatings with different nickel network densities are shown in Figure 2B. The value of the surface resistance markedly decreased with an increase in the surface density of the network. For the highest coating density, R_s_ was 180 Ω/sq. However, it should be noted that as the coating density approached the value of 200 µg/cm^2^, the relative error in measuring the R_s_ values increased significantly. This can be explained by an increase in the heterogeneity of the submicron network deposition since the magnetization of the entire layer increased. In the multistage deposition of submicron fibers, the magnetization of the underlying layer caused the reorientation and disorientation of the upper layers before they had time to fix. Table 1 shows the FoM quality index values calculated using Formulas (1) and (2) (FoM1 and FoM2) for the several samples studied.

Thus, due to the studies carried out, the dependences of the transparency coefficient T550 and surface resistance Rs of the developed electrically conductive coating on the surface density of nickel networks on a glass substrate were obtained. The most optimal density values corresponding to the maximum values of the FoM quality index were established. It is interesting to study the possibility of increasing the electrical conductivity of the developed material by modifying the PEDOT: PSS polymer matrix. The authors in [93] showed that the electrical conductivity of the PEDOT: PSS film significantly improved (by three orders of magnitude) compared with the conductivity of the original film, after doping the initial PEDOT: PSS dispersion with p-toluenesulfonic acid (PTSA) and the post-treatment of the finished film with dimethyl sulfoxide (DMSO). Poly(3,4-ethylenedioxythiophene) chains were oxidized via PTSA. This led to the formation of positively charged holes in the thiophene chain of poly(3,4-ethylenedioxythiophene) (PEDOT), which could contribute to the conduction. The polymer matrix was modified by adding PTSA to the finished PEDOT: PSS aqueous dispersion. The concentration of PTSA varied in the range of 0.25–1 wt.% in the final dispersion. The higher the acid concentration in the aqueous dispersion, the higher the conductivity of the resulting polymer film [93]. However, it is worth remembering that the viscosity of the solution also increased. This fact made it difficult to apply to the surface and led to the formation of inhomogeneities.

In general, the technique for designing an electrically conductive coating on the glass is as follows: (1) The first step involved the synthesis of a system of oriented nickel nanonetworks (nanonetworks) combined into a single system by an oriented network of nickel submicron fibers on the glass surface. Previously, it was shown that the optimal value of the surface density of nickel is 218 µg/cm^2^. This amount of nickel was used in all the subsequent experiments. (2) An aqueous dispersion of PEDOT: PSS (3 wt.%) modified with p-toluenesulfonic acid was applied onto a system of nickel networks using a spin coater. As a result of this stage, we obtained a modified version of the developed coating {Ni + PEDOT: PSS + PTSA}. First of all, an aqueous solution was prepared with p-toluenesulfonic acid (monohydrate) 50 wt.%. Acid monohydrate is very soluble in water. We used PTSA monohydrate (>98.5%) Sigma-Aldrich. Further, this acid solution was added to an aqueous dispersion of PEDOT: PSS (3 wt.%) in different proportions to obtain the required concentration in the range of 0.25–1 wt.%. The finished mixture was stirred with a magnetic stirrer for 2 h. The result was a fairly thick liquid. The resulting {PEDOT:PSS + PTSA} solutions were then applied via spin-coating with a fixed program: 1000 RPM (15 s)–2000 RPM (45 s)–3000 RPM (15 s). The applied film was dried at a temperature of 120 °C for 20 min. It is known from the literature that an excessive amount of polystyrene sulfonate components is formed on the surface of the PEDOT: PSS film, which leads to the formation of an insulating layer [93,94,95]. The application of polar-solvent vapor annealing (the so-called PSVA technique) to PEDOT: PSS films causes significant segregation of excess polystyrene sulfonate on the film surface, which leads to an improvement in the conductive channel between the PEDOT regions, thereby increasing the conductivity of the film as a whole. Therefore, the finished dried film was additionally annealed in DMSO vapor (>99.5%, Sigma-Aldrich) for 30 min at a temperature of 130–140 °C. The final step was washing the resulting layer with isopropyl alcohol and deionized water and drying it at a temperature of 120 °C. We investigated the surface of the obtained coatings via atomic force microscopy (AFM). A significant change in the surface topography of the obtained coating occurred after doping PEDOT:PSS with p-toluenesulfonic acid (Figure 3).

Coating {nickel network system + unmodified PEDOT:PSS} was used as a control. Compared with the control sample, the surface roughness (R_a_ value) greatly increased, and the surface became more rough and uneven. For the modified surface, the R_a_ value ranged from 160 to 180 nm for different concentrations of PTSA in the aqueous dispersion (Figure 4), while the R_a_ value was approximately 6 nm for the unmodified coating. The formation of large associates caused this change with sizes of 2–5 μm on the surface due to doping PEDOT: PSS with p-toluenesulfonic acid.

An overview of the surface of the synthesized coating is shown in Figure 5A. Film inhomogeneities mainly manifested themselves in topography, i.e., changes in the relief of the film surface. When studying through optical and scanning electron microscopy, no noticeable contribution from the associates detected using AFM was found. Scanning electron microscopy, among other advantages, indirectly made it possible to observe the uniformity of the electrically conductive layer distribution. A negative charge accumulated in an isolated nonconductive region, which could not “drain” from the sample’s surface (film or coating), thereby causing distortions in the form of a glow in the SEM images. The areas free from nickel networks are evenly contrasted against the general background of the image (Figure 5B). This fact indirectly confirms the presence of a unifying electrically conductive component of the synthesized composite material.

The change in the electrically conductive properties of the modified coating is shown in Figure 6A. The coating surface resistance R_s_ corresponding to different concentrations of PTSA in an aqueous dispersion of PEDOT: PSS was studied using the van der Pauw four-probe method.

We have shown that doping PEDOT: PSS with p-toluenesulfonic acid leads to a significant decrease in the surface resistance of the previously developed optically transparent electrically conductive coatings. On average, it was possible to reduce the surface resistance value by almost eight times. An increase in the concentration of p-toluenesulfonic acid in an aqueous dispersion of PEDOT: PSS from 0% to 0.5% led to a monotonic decrease in the value of the surface resistance of the resulting material. A further increase in the PTSA concentration did not give a stable result in increasing the surface conductivity, although such a trend exists for a pure polymer film [93]. Experimental data (Figure 6A) showed a slight upward bend in the curve connecting the points. For a PTSA concentration value of 1 wt.%, a slight increase in the surface resistance value was observed. This fact is inconsistent with the literature data corresponding to a pure Pedot:PSS polymer film [93]. We assume that this is due to a significant increase in the viscosity of the Pedot:PSS + PTSA (1 wt.%) solution compared with less concentrated ones. This led to the formation of inhomogeneities in the polymer film. The process of applying the polymer becomes more complicated with an increase in the concentration of PTSA because the solution becomes very viscous, which is also noted in [93]. The resulting value of R_S_ remained approximately at the same level in our case. We hypothesize that this is because electrical conductivity is largely determined by the metal network rather than the polymer matrix. To determine the type of conductivity of the synthesized coating, we studied the temperature dependence of the surface resistance value (Figure 7).

In the studied temperature range, the dependence monotonically increased, and all experimental points lay well on a straight line. The linearly increasing dependence was in good agreement with the metallic conductivity model. The experimental data deviated from a linear dependence for pure conductive polymers, which is approximated by the following exponential form [96,97,98,99]:(3)$$\rho \left(T\right)=\rho_{0}exp{\left(-T_{0}/T\right)}^{\alpha }$$

The degree α can take on different values depending on the properties of a particular polymer. For PEDOT: PSS, it has been experimentally established that α has a value close to 0.5 [96]. Therefore, we assume that the nickel–metal network largely determined the conductivity of the coating we developed.

Along with the change in the electrical properties of the modified coating, we found a difference in the layer’s optical properties (Appendix A). An increase in the concentration of p-toluenesulfonic acid in an aqueous dispersion of PEDOT: PSS led to a decrease in the transparency coefficient of the resulting coating over the entire optical range. As a numerical characteristic of transparency in the visible range, the value of the transparency coefficient at a wavelength of 550 nm is usually chosen. As the concentration of PTSA increased, the value of T_550_ decreased monotonically (Figure 6B).

To determine the optimal parameters for the synthesis of an optically transparent electrically conductive coating and the advisability of doping PEDOT: PSS with p-toluenesulfonic acid, we used the figure of merit (FoM) numerical value. Previously, we used two variants of its calculation: FoM1 and FoM2 [85,86]. The calculated values of these parameters for the studied series of samples are presented in Table 2.

The highest FoM1 value corresponded to the 0.5% PTSA coating. However, the calculated FoM2 value was higher at 0.75%. It can be seen that the calculation formula for FoM2 gave very close values for both 0.5% and 0.75%. From our point of view, the use of 0.5% PTSA for doping PEDOT: PSS was more rational because such a solution was much easier to apply via spin-coating onto a solid surface.

A comparison was made with several similar works studying the modification of the PEDOT:PSS polymer to verify the obtained results. In [101], two methods of PEDOT:PSS polymer treatment were considered: perfluorooctane sulfonic acid (PFOSA) and sulfuric acid. In the first case, a reduction in Rs to about 100 ohm/sq was demonstrated, and in the second case, a reduction to about 200 ohm/sq was observed. In this research, the transparency coefficient was close to the value of 75–80%, which is comparable with the material developed by us at a noticeably higher value of Rs. The authors of [101] proposed a film treatment with formic acid to enhance the conductivity of PEDOT:PSS. A good result was achieved; specifically, high transparency with good conductivity (92.1% with 145 ohm/sq sheet resistance) was demonstrated. This result exceeds our results in terms of transparency coefficient but is inferior in terms of electrical properties. It should be noted that the material developed by us based on oriented nickel networks has much more stable electrical characteristics. This is ensured by the presence of a metal frame in the polymer matrix. A method to fabricate multilayered thin films with higher-ordered structures was developed in [102]. The etching process with H2SO4 and dimethyl sulfoxide (DMSO) was used to improve its electrical properties. The thickness of PEDOT:PSS thin films was experimentally optimized to maximize the enhancement of carrier mobility via a layer-by-layer (LBL) process. The combined method consisting of etching and the LBL process showed improvement in the charge carrier mobility. However, only electrical conductivity was studied in that work. The authors achieved a value of 50 ohms/sq at a thickness of 110 nm. For comparison, the coating we developed had a value of 35 ohm/sq at a thickness of 22 nm. We did not consider transparency in the optical range, which, for even pristine PEDOT:PSS, can be very low at a thickness of 110 nm. In the considered works [100,101,102], a serious decrease in the electrical conductivity of the material was observed during the first month. The composite material developed in our work showed significantly more stable results even during a year after development (Figure 8).

## 4. Conclusions

We showed that the doping of PEDOT: PSS with p-toluenesulfonic acid in designing an optically transparent electrically conductive composite coating based on oriented nickel networks in a polymer matrix is expedient. It was established that adding p-toluenesulfonic acid to an aqueous dispersion of PEDOT: PSS with a concentration of 0.5% led to a decrease in the surface resistance of the resulting coating by eight times. The reduction in the transparency coefficient at a wavelength of 550 nm for the resulting layer was insignificant. In addition, we studied the dynamics of the optical and electrical properties of the developed material for one year: changes in the values of surface resistance and transparency coefficient did not exceed the measurement error.

## Figures and Tables

**Figure 1 nanomaterials-13-00831-f001:**
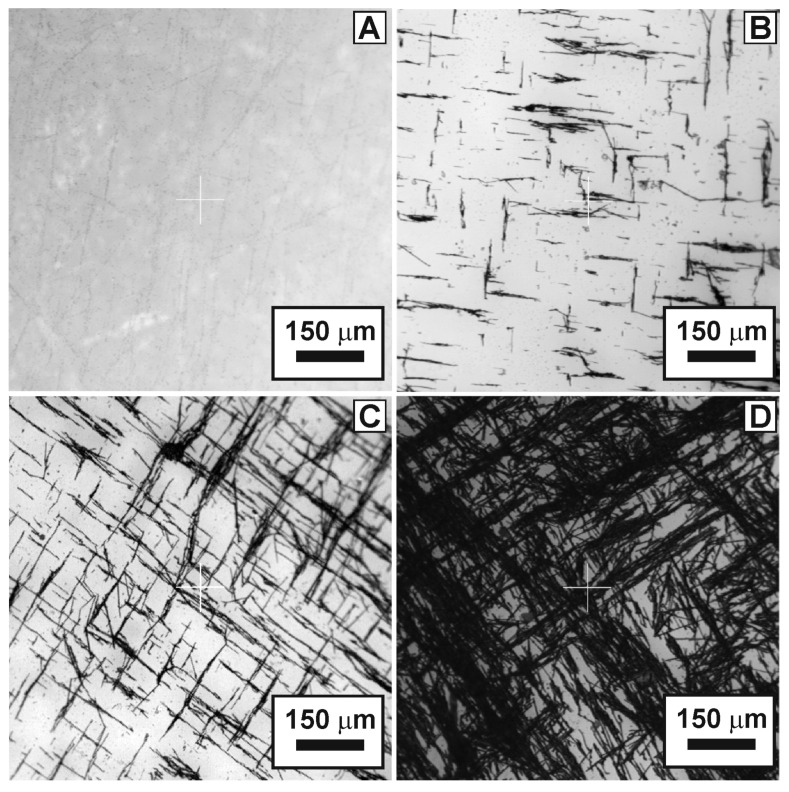
Optical images of coatings with nickel networks on the surface of a glass substrate with different densities: 0.6 μg/cm^2^ (**A**), 31 μg/cm^2^ (**B**), 62 μg/cm^2^ (**C**), and 311 μg/cm^2^ (**D**).

**Figure 2 nanomaterials-13-00831-f002:**
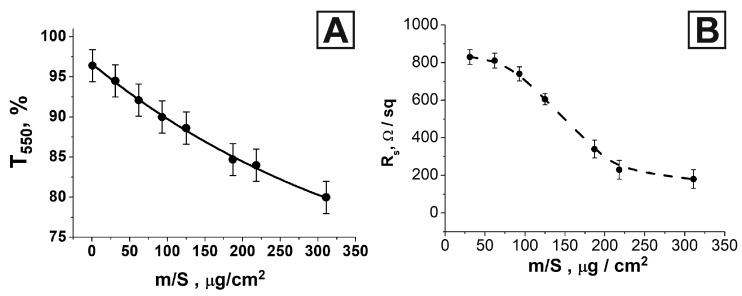
Dependence of the transparency coefficient at a wavelength of 550 nm on the deposited nickel amount on a glass substrate (**A**); dependence of the surface resistance on the amount of deposited nickel on the glass surface (**B**).

**Figure 3 nanomaterials-13-00831-f003:**
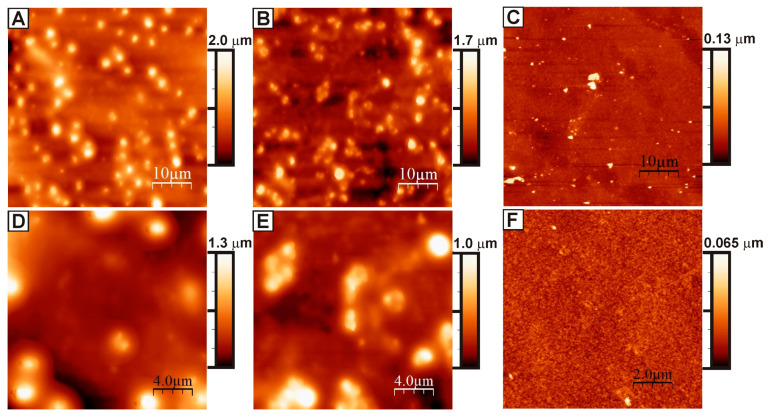
AFM images of the surface of the obtained coatings {Ni + PEDOT:PSS + PTSA}: the concentration of PTSA in the aqueous dispersion was 0.75 wt.% (**A**,**D**) and 1.00 wt.% (**B**,**E**); pristine PEDOT: PSS (**C**,**F**).

**Figure 4 nanomaterials-13-00831-f004:**
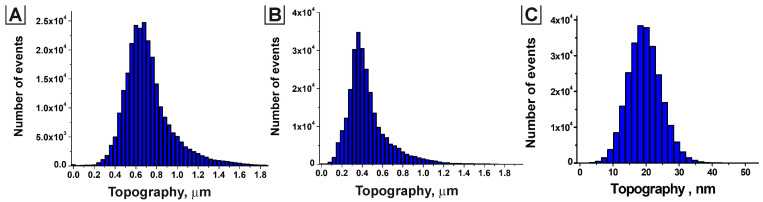
Topographic histograms corresponding to PTSA concentration in the aqueous dispersion of 0.75 wt.% (**A**) and 1.00 wt.% (**B**); pristine PEDOT: PSS (**C**).

**Figure 5 nanomaterials-13-00831-f005:**
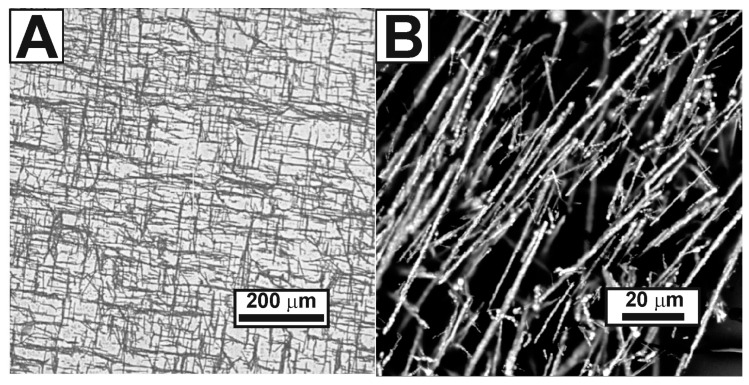
Images of the resulting coating surface {Ni + PEDOT: PSS + PTSA} (PTSA concentration in the aqueous dispersion is 0.75 wt.%): images obtained with an optical microscope (**A**) and a scanning electron microscope (**B**).

**Figure 6 nanomaterials-13-00831-f006:**
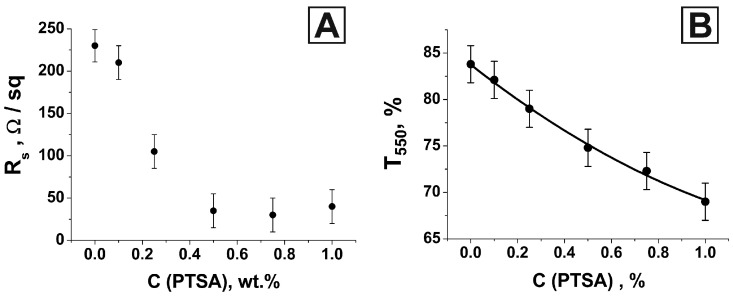
Dependence of the surface resistance of the resulting layer on the concentration of PTSA in an aqueous dispersion of PEDOT: PSS (**A**); dependence of the transparency coefficient of the modified {Ni + PEDOT: PSS + PTSA} coating on a glass substrate at a wavelength of 550 nm on the concentrations of PTSA in the initial aqueous dispersion of PEDOT: PSS (**B**).

**Figure 7 nanomaterials-13-00831-f007:**
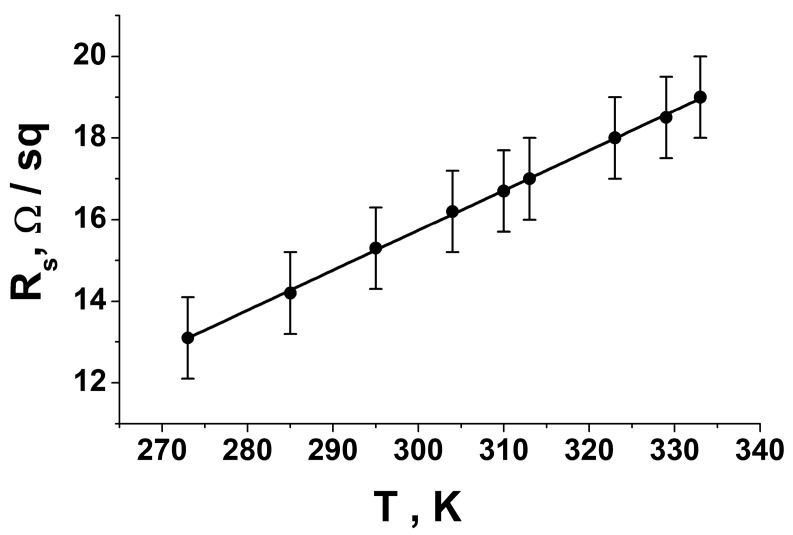
Temperature dependence of the surface resistance value of the developed coating.

**Figure 8 nanomaterials-13-00831-f008:**
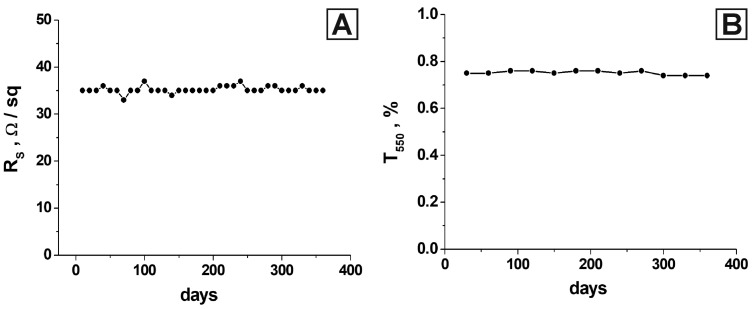
The stability of the obtained material’s parameters (PTSA concentration of 0.5 wt.%): surface resistance (**A**) and transparency coefficient at 550 nm (**B**).

**Table 1 nanomaterials-13-00831-t001:** Main characteristics of the developed electrically conductive transparent coating at different nickel deposition densities.

m/S, 10^−6^ g/cm^2^	R_s_, Ohm/sq	T_550_	FoM1, 10^−4^ Ohm^−1^	FoM2
31	830	0.94	6.5	7.2
62	810	0.92	5.4	5.5
93	740	0.90	4.7	4.7
125	605	0.89	5.2	5.2
187	340	0.85	5.8	6.6
218	230	0.84	8.0	9.0
311	180	0.80	5.9	8.9

**Table 2 nanomaterials-13-00831-t002:** The main characteristics of the developed electrically conductive transparent coating corresponding to different concentrations of PTSA in an aqueous dispersion of PEDOT: PSS.

C (PTSA), wt.%	R_s_, Ohm/sq	T_550_	FoM1, 10^−4^ Ohm^−1^	FoM2
0	230	0.84	8.0	9.0
0.1	210	0.82	6.5	8.6
0.25	105	0.79	9.0	14.4
0.5	35	0.75	16.1	34.8
0.75	30	0.72	12.5	35.2
1.0	40	0.69	6.1	23.1
ITO [100]	7	0.87	354.9	373.4

## Data Availability

Not applicable.

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
