# Peer review of "Doping of Transparent Electrode Based on Oriented Networks of Nickel in Poly(3,4-Ethylenedioxythiophene) Polystyrene Sulfonate Matrix with P-Toluenesulfonic Acid"

_nanomaterials, 2023, doi:10.3390/nano13050831_

Round 1

Reviewer 1 Report

The present work  obtains an optically transparent electrode based on oriented 12 nanonetworks of nickel in poly(3,4-ethylenedioxythiophene) polystyrene sulfonate matrix. Some minor issues should be addressed before publication:

(1) Some background about the study are suggested to be added to the Abstract;

(2) Page 6, Line241-243, "As expected, with an increase in the density of nickel deposition on the glass surface, a monotonous decrease in the transparency coefficient is observed over the entire optical range.", some more detailed description and explanations are suggested to be added;

(3) Page 6, Line 250-251, "This phenomenon was discussed in [95]", the specific discussion is suggested to be described again in the present work;

(4) Page 17, Ref [100]-[102], what is the meaning of “E2, E3, E4, E5”?

(5) Page 11, Line 373-375, "A further increase in the PTSA concentration did not give a stable result in increasing the surface conductivity, although such a trend exists for a pure polymer film", could the author give some detailed explanations?

Overall, the paper is well organised and can be accepted after revision.

Author Response

Point 1: Some background about the study are suggested to be added to the Abstract;

Response 1: Abstract has been modified. The following sentences have been added to the Abstract:

“Optically transparent electrodes are used in many modern devices. Therefore, the search for new inexpensive and environmentally friendly materials for them remains an urgent task. We have previously developed a material for optically transparent electrodes based on oriented platinum nanonetworks. The technique has been upgraded to obtain a cheaper option from oriented nickel networks.”

Point 2: Page 6, Line241-243, "As expected, with an increase in the density of nickel deposition on the glass surface, a monotonous decrease in the transparency coefficient is observed over the entire optical range.", some more detailed description and explanations are suggested to be added;

Response 2: The manuscript has been corrected. The following sentences have been added to the manuscript:

“With an increase in the surface density of nickel on glass, a decrease in the total area of uncoated (transparent) areas is observed (Figure 1). Therefore, one should expect a decrease in the transparency coefficient in the entire optical range for denser coatings compared to less dense ones. Transmission spectra of oriented nickel networks on a glass substrate in the UV, visible, and near-IR regions for different amounts of deposited metal are shown in Figure S7.”

Point 3: Page 6, Line 250-251, "This phenomenon was discussed in [95]", the specific discussion is suggested to be described again in the present work;

Response 3: The manuscript has been corrected. The following sentences have been added to the manuscript:

“The phenomenon can be related to the peculiarities of radiation transmission through subwavelength apertures. Bethe's criterion shows that when the aperture size is less than half the wavelength of the radiation passing through it, its strong attenuation is observed already in the near field. In our case, this means that the material's absorption coefficient increases when the critical value of the wavelength is reached. It should be noted that such an effect is possible only for regular structures, in which the spread of transverse dimensions is small. We believe that the discovered effect may be of interest for applications in the development of nanosensors and near-infrared sensors.”

Point 4: Page 17, Ref [100]-[102], what is the meaning of “E2, E3, E4, E5”?

Response 4: It's a typo. The references have been corrected.

Point 5: Page 11, Line 373-375, "A further increase in the PTSA concentration did not give a stable result in increasing the surface conductivity, although such a trend exists for a pure polymer film", could the author give some detailed explanations?

Response 5: The manuscript has been corrected. The following sentences have been added to the manuscript:

“Experimental data (Fig.6A) show a slight upward bend in the curve connecting the points. For a PTSA concentration value of 1 wt.%, a slight increase in the surface resistance value is observed. This fact is inconsistent with the literature data corresponding to a pure Pedot:PSS polymer film [96]. We assume that this is due to a significant increase in the viscosity of the Pedot:PSS + PTSA (1 wt.%) solution compared to less concentrated ones. This leads to the formation of inhomogeneities in the polymer film.”

Reviewer 2 Report

In this article,  authors have studied the nickel networks with modified PEDOT:PSS as a transparent electrode and systematically  investigated the conductivity and transparencies of the thin films. Study might be helpful to other researchers working on the topic. However, reviewer recommends to address following comments before considering for the publication

·         In the introduction at the end add a paragraph on what you are studying in present work

·         Add full names of all chemical/compounds in the procedures

·          Explain the role of the magnetic field in detail in the results section, also it is making the process more complex than other studies where only spin coating is used.  

·         Fig. S7 labelling and on the graphs is confusing with the colors or is either incorrect

·         As authors have seen from Fig. S9 and Fig S6, that transparency is significantly reduced. also it increases the roughness. so, what advantage it will bring for the optoelectronic devices as it will substantially reduce the light in-coupling or outcoupling from the device. So, what are the advantages even if  it increases the conductivity

·         The PEDOT:PSS has already been investigated in detail by many previous works, even for ITO free electrodes just by PEDOT:PSS modifications, and commercially highly conductive PEDOT:PSS is available. So what are the benefits of using with the nickel networks

https://pubs.acs.org/doi/10.1021/am405024d

https://www.sciencedirect.com/science/article/pii/S1566119919303283

https://www.mdpi.com/2079-4991/10/11/2211

·         It is recommended to add the conductivity and transparencies of the ITO as a reference, in their studies

·         Fig 3 and Line 338 and please add the unmodified film images

·         In the conclusion author mentions about the stability for 1 year, please add the data

·         If possible to do, please use your electrodes in the actual optoelectronics device

Author Response

Point 1: In the introduction at the end add a paragraph on what you are studying in present work.

Response 1: The manuscript has been corrected. The following sentences have been added to the “Introduction”:

“In the research we have studied the nickel networks with modified PEDOT:PSS as a transparent electrode. A thin coating is created on the surface of a glass substrate according to a technique developed earlier and described in [85]. In general, the work is a logical continuation of the investigation [85] with the improvement of the method and detailed research on the influence of process parameters on the synthesized material.”

Point 2: Add full names of all chemical/compounds in the procedures.

Response 2: Full names of all chemical compounds have been added in the “Materials and Methods” section of the manuscript.

Point 3: Explain the role of the magnetic field in detail in the results section, also it is making the process more complex than other studies where only spin coating is used.

Response 3: The manuscript has been corrected. The following sentences have been added:

“The use of a CTAB micellar template leads to the formation of metal nano-networks with a domain structure of orientation: the orientation of the network changes from a domain to a domain [85].  Domain sizes range from one hundred microns to a millimeter in transverse dimension. The sizes of domains are presumably determined by the formation of a micellar template at the water-glass interface. Nickel is an inexpensive metal that is magnetized. This phenomenon can be used to control its shape and direction of growth. We applied a homogeneous magnetic field to the area of nanonetwork synthesis so that its field lines were parallel to the glass surface. The use of an external magnetic field makes it possible to obtain a long-range order orientation and to single out the preferred direction of the domain orientation as a whole.”

Point 4: Fig. S7 labelling and on the graphs is confusing with the colors or is either incorrect

Response 4: The legend of Fig. S7 was incorrect. Figure S7 has been corrected and added to Supplementary.

Point 5: As authors have seen from Fig. S9 and Fig S6, that transparency is significantly reduced. also it increases the roughness. so, what advantage it will bring for the optoelectronic devices as it will substantially reduce the light in-coupling or outcoupling from the device. So, what are the advantages even if it increases the conductivity.

Response 5: This is a fair remark. Transparency is significantly reduced. This is a cumulative effect associated with both an increase in roughness and an increase in the optical density of the material. Indeed, it is quite difficult to correctly assess the quality of a transparent electrode using only one of the parameters RS or T. That is why researchers use a function that depends on both parameters at once. In our work, we also use such quality assessment functions FoM1 and FoM2, which allow us to compare different materials within the stated topic. These functions allow independent scientific groups to compare their results with each other. In our case, the obtained parameters show a noticeable increase in the FoM1 and FoM2 values. This confirms the feasibility of doping with p-toluenesulfonic acid.

Point 6: The PEDOT:PSS has already been investigated in detail by many previous works, even for ITO free electrodes just by PEDOT:PSS modifications, and commercially highly conductive PEDOT:PSS is available. So what are the benefits of using with the nickel networks.

Response 6: The manuscript has been corrected. The following sentences have been added:

“A comparison can be made with several similar works studying the modification of the PEDOT:PSS polymer to determine the level of results obtained. In [103], two methods of PEDOT:PSS polymer treatment are considered: perfluorooctane sulfonic acid (PFOSA) and sulfuric acid. In the first case, a reduction in Rs to about 100 ohm/sq is demonstrated, and in the second case, to about 200 ohm/sq. In this research, the transparency coefficient is close to the value of 75-80%, which is comparable with the material developed by us at a noticeably higher value of Rs. Authors of work [104] proposed a film treatment with formic acid to enhance the conductivity of PEDOT:PSS. A good result was achieved: high transparency with good conductivity (92.1% with 145 ohm/sq sheet resistance) is demonstrated. This result exceeds ours in terms of transparency coefficient but is inferior in terms of electrical properties. It should be noted that the material developed by us based on oriented nickel networks has much more stable electrical characteristics. This is ensured by the presence of a metal frame in the polymer matrix. A method to fabricate multi-layered thin films with higher-ordered structures was developed in [105]. The etching process with H2SO4 and dimethyl sulfoxide (DMSO) is used to improve its electrical properties. The thickness of PEDOT:PSS thin films was experimentally optimized to maximize the enhancement of carrier mobility via a layer-by-layer (LBL) process. The combined method consisting of etching and the LBL process showed the improvement of the charge carrier mobility. However, only electrical conductivity is studied in the work. The authors achieved a value of 50 ohms/sq at a thickness of 110 nm. For comparison, the coating we developed has a value of 35 ohm/sq at a thickness of 22 nm. The paper does not consider transparency in the optical range, which for even pristine PEDOT:PSS can be very low at a thickness of 110 nm. In the considered works [103-105] a serious decrease in the electrical conductivity of the material is observed during the first month. The composite material developed in our work shows significantly more stable results even during the year (Figure 8).”

Point 7: It is recommended to add the conductivity and transparencies of the ITO as a reference, in their studies.

Response 7: The conductivity and transparency of the ITO have been added to the manuscript (Table 2).

Point 8: Fig 3 and Line 338 and please add the unmodified film images.

Response 8: The unmodified film images have been added to Figure 3. Figure 4 has been corrected.

Point 9: In the conclusion author mentions about the stability for 1 year, please add the data.

Response 9: Figure 8 (the stability for 1 year) has been added to the manuscript.

Point 10: If possible to do, please use your electrodes in the actual optoelectronics device.

Response 10: At present, we do not have the ability to conduct tests in optoelectronic devices, unfortunately. This requires a noticeable reorientation into the field of applied engineering, which is not very easy for us at this stage. However, we hope to delve into this direction in the next phase of the work.
